# National adaptation and implementation of WHO Model List of Essential Medicines: A qualitative evidence synthesis

**Elizabeth F. Peacocke**[1]\*, **Sonja L. Myhre**[1], **Hakan Safaralilo Foss**[2],
**Unni Gopinathan**[1]

**1** Global Health, Division for Health Services, Norwegian Institute of Public Health, Oslo, Norway, **2** Faculty of Medicine, University of Oslo, Oslo, Norway

\* Elizabeth.Peacocke@fhi.no

## Abstract

**Data Availability Statement:** All relevant data are within the manuscript and its Supporting Information files.

### Background

The World Health Organization Model List of Essential Medicines (WHO EML) has played a critical role in guiding the country-level selection and financing of medicines for more than 4 decades. It continues to be a relevant evidence-based policy that can support universal health coverage (UHC) and access to essential medicines. The objective of this review was to identify factors affecting adaptation and implementation of WHO EML at the national level.

### Methods and findings

We conducted a qualitative evidence synthesis by searching 10 databases (including CINAHL, Embase, Ovid MEDLINE, Scopus, and Web of Science) through October 2021. Primary qualitative studies focused on country-level implementation of WHO EML were included. The qualitative findings were populated in the Supporting the Use of Research Evidence (SURE) framework, and key themes were identified through an iterative process. We appraised the papers using the Critical Appraisal Skills Programme (CASP) tool and assessed our confidence in the findings using the Grading of Recommendations Assessment, Development and Evaluation working group-Confidence in Evidence from Reviews of Qualitative research (GRADE-CERQual). We screened 1,567 unique citations, reviewed 183 full texts, and included 23 studies, from 30 settings. Non-English studies and experiences and perceptions of stakeholders published in gray literature were not collected.

Our findings centered around 3 main ideas pertaining to national adaptation and implementation of WHO EML: (1) the importance of designing institutions, governance, and leadership for national medicines lists (NMLs), particularly the consideration of transparency, coordination capacity, legislative mechanisms, managing regional differences, and clinical guidance; (2) the capacity to manage evidence to inform NML updates, including processes for contextualizing global evidence, utilizing local data and expert knowledge, and assessing budget impact, to which locally relevant cost-effectiveness information plays an important

**Funding:** This work was partly supported by the Research Council of Norway through the Global Health and Vaccination Programme (GLOBVAC Project 234608). The funder had no role in the study design, data collection and analysis, decision to publish, or preparation of the manuscript.

**Competing interests:** The authors have declared that no competing interests exist.

**Abbreviations:** CASP, Critical Appraisal Skills Programme; GRADE-CERQual, Grading of Recommendations Assessment, Development and Evaluation working group-Confidence in Evidence from Reviews of Qualitative research; HIC, high-income country; HTA, Health Technology Assessment; LMIC, low- and middle-income country; NML, national medicine list; PRISMA, Preferred Reporting Items for Systematic Reviews and Meta-Analyses; SDG, Sustainable Development Goal; STG, standard treatment guideline; SURE, Supporting the Use of Research Evidence; UHC, universal health coverage; WHO EML, World Health Organization Model List of Essential Medicines.

role; and (3) the influence of NML on purchasing and prescribing by altering provider incentives, through linkages to systems for financing and procurement and donor influence.

## Conclusions

This qualitative evidence synthesis underscores the complexity and interdependencies inherent to implementation of WHO EML. To maximize the value of NMLs, greater investments should be made in processes and institutions that are needed to support various stages of the implementation pathway from global norms to adjusting prescribed behavior. Moreover, further research on linkages between NMLs, procurement, and the availability of medicines will provide additional insight into optimal NML implementation.

## Protocol registry

PROSPERO CRD42018104112

---

## Author summary

### Why was this study done?

- The World Health Organization Model List of Essential Medicines (WHO EML) has played a critical role in guiding the country-level selection and financing of medicines for more than 4 decades.

- National medicine lists (NMLs) are perceived to be an important part of a country's medicines policy; however, few efforts have been made to systematically integrate insights of WHO EML implementation from the empirical literature.

- The objective of this study was to identify the factors that influence implementation of global normative guidance on essential medicines and provide insight on areas where additional support may facilitate country-level implementation.

### What did the researchers do and find?

- A qualitative evidence synthesis was undertaken, using the GRADE-Confidence in the Evidence from Reviews of Qualitative research (CERQual) approach to assess how much confidence to place in the findings. A systematic search of 10 databases identified 23 articles for inclusion after assessing 1,567 unique citations and reviewing 183 full texts.

- We found that implementation can be facilitated by national medicine selection committees that operate with consultative mandates, clear leadership and oversight, and monitoring and evaluation.

- Implementation of NMLs also requires harmonization with reimbursement processes and recommended clinical practice. National standard treatment guidelines (STGs) therefore play a crucial role in translating intentions of NML to clinical practice, while legislation, oversight, and monitoring are additional tools for ensuring compliance.

- The types of information used in adaptations are extremely important. Crucial to country relevant updates of NMLs is the balancing of global evidence, expert knowledge, and local data, and there is an opportunity for further use of health economic methods to inform decision-making on essential medicines.

### What do these findings mean?

- Updating NMLs following biannual global revisions of WHO EML requires significant financial and human resource investment by countries. The number of actors and processes underscore the complexity and interdependencies inherent to implementation of the EML.

- These findings suggest that to maximize the value of NMLs, greater investments should be made in different types of institutions that are needed to support various stages along the implementation pathway from global norms to adjusting prescriber behavior.

## Introduction

The World Health Organization Model List of Essential Medicines (WHO EML) offers a global evidence–informed reference list for countries to use in the adaptation of their national formulary or national medicines list (NMLs). It was established on the premise that some medicines are more important than others and hence should be defined as "essential" and that access to these should be strengthened [1,2]. The original 1977 list included 186 medicines deemed essential for every healthcare system and provided guidance on medicine selection for NMLs. The scope of WHO EML Expert Committee on Selection and Use of Medicines is to consider medicines from the perspective of public health relevance, evidence on efficacy and safety, and comparative cost-effectiveness to identify those medicines that best satisfy the priority healthcare needs of a country's population [3].

   Currently, there are 2 WHO EMLs: WHO Essential Medicines List and WHO Essential Medicines List for Children (first created in 2007, from here on referred to as WHO Children's List), and both are revised every 2 years. The categorization of medicines is twofold: "core" medicines represent the minimum medicines needed, while "complementary" medicines require specialized medical care, diagnostic, or monitoring facilities. From 2002, WHO sought to institutionalize an evidence-based process for making decisions about the inclusion of new medicines to WHO EML [2].

   WHO recommends that all countries formulate and implement a comprehensive national medicines policy to improve access to safe and effective medicines of good quality [4]. A country's NML is a government-approved list of medicines, which can often be adapted and implemented as a local formulary, or as a secondary list to the NML "tertiary list" [5]. The NML is intended to guide public sector procurement and supply, reimbursement schemes, medicine donations, and local production [6]. Moreover, NMLs can aid countries to prioritize medicines and can be used as the foundation for reimbursement schemes and national treatment guidelines (standard treatment guidelines, STG). Medicine use is a key driver of healthcare expenditure; thus, implementing a NML can be a strategy for promoting efficient use of healthcare resources [7]. However, in many contexts, the inclusion of medicines on the NML

does not necessarily guarantee that these are accessible to populations, for example, due to stockouts or high out-of-pocket costs. The implementation process of a NML, including the steps for making these medicines accessible [8,9], is therefore more complex than a binary decision to include or exclude a medicine on the list.

WHO EML is primarily used by countries as a basis for guiding national decisions about their own NMLs [8,9]. For several reasons, it has been heavily debated whether WHO EML or WHO Children's List serve as an optimal point of reference for national medicines policy. A key issue has been that WHO EML is meant to define minimum needs for a health system and therefore do not necessarily include all effective medicines that may be necessary for a country. For example, it was only in 2002 that antiretroviral drugs against HIV/AIDS were included on the list, in spite of the increasing severity and destabilizing effect of the HIV/AIDS epidemic [10]. The delayed inclusion of antiretroviral medicines reflected the fact that affordability until then was a precondition for selection into WHO EML [11]. Since then, affordability has been viewed as a consequence that must be managed after selection into the list [12]. The additions of high-cost medicines for cancers, hepatitis C, and multidrug-resistant tuberculosis reinforced this way of dealing with affordability of included medicines [12]. In some cases, medicines on WHO EML have remained despite most nations using better and more cost-effective options [1]. Another tension has been between WHO EML and lack of consistency with treatment guidelines issued by other WHO committees [1], as well as discrepancies between national treatment guidelines and the medicines on the EML. In the latter case, there was a relatively long and notable gap between modern clinical practice guidelines for preventing and treating cardiovascular diseases and the medicines included on the list [13]. However, with concerted efforts, WHO EML has been modernized over time [13,14]. Questions have also been raised about the standard of applications submitted to WHO Expert Committee for Essential Medicines and the transparency of their decision-making process [15], which, arguably, has improved since major reforms were implemented in 2001 [12]. Despite these contentious issues, individual studies from a wide range of settings underscore that WHO EML often is a starting point for national medicines selection processes, particularly in low- and middle-income countries (LMICs) [16–20].

In response to global normative guidance that WHO EML represents, countries seek to adapt it by considering factors such as the disease burden in the country, the cost of medicines, specific concerns of patients or providers, and health systems capacity to deliver medicines to patients. In addition, other country-level considerations may influence this process such as the demographic profile, climate, and transportation infrastructure. To assess these factors, countries might establish institutions such as standing committees, set up processes for producing evidence reviews to inform cost-effectiveness, and assess preferences of patients and providers [21,22]. The revision of WHO EML is not accompanied by detailed guidance for national medicines selection processes [16]. Many articles have been published on this topic, especially studies that compare the coverage and access of medicines as listed on WHO EML with a country's NML. However, the authors are not aware of any systematic efforts to analyze country-level experiences with adapting and implementing essential medicines lists. The implementation of WHO EML involves many steps that need to be better understood, resourced, and executed [19]. Thus, there is a need to explore the theory practice gap, namely how the global norm on essential medicines is integrated in real-world policymaking and translated to value for prescribers and patients.

Four decades since its launch, WHO EML continues to play an important role internationally. It is intertwined in the United Nations Sustainable Development Goal (SDG) on global health and well-being, specifically, SDG targets 3.8 (universal health coverage, UHC) and 3b (access to medicines for all) [23]. By systematically integrating insights from qualitative studies

of WHO EML implementation, we hope to better understand what impedes or facilitates adapting global normative guidance on essential medicines at the country level and what processes need to be in place to optimize implementation. Accordingly, the primary objective of this review was to identify factors affecting adaptation and implementation of WHO EML at the national level.

## Methods

### Study design

A qualitative evidence synthesis [24] approach was chosen to synthesize evidence about the implementation of WHO EML and WHO Children's List. This study is reported as per the Preferred Reporting Items for Systematic Reviews and Meta-Analyses (PRISMA) guideline (S1 Checklist). Throughout the review, we were also guided by Arksey and O'Malley's 5-stage framework to secure a transparent process, enable replication of the search strategy, and increase the reliability of the study findings [25,26]. A study protocol was drafted and updated during our review and was published on the PROSPERO international prospective register for systematic reviews [27].

### Key definitions and theory

In this study, the adaptation and implementation of WHO EML was defined as the transfer and use of the global norm at the country level, namely through a NML. We built our understanding of WHO EML implementation on well-established theoretical frameworks outlining key steps of the policymaking life cycle, which include agenda setting, policy formulation, policy adoption, policy implementation, and policy evaluation [28–31]. Our inquiry focused on policy adoption and policy implementation [31]. Policy adoption involves the formal adoption of a policy solution, usually by politicians, policymakers, or bureaucrats in government, which favors a specific solution or strategy for addressing the problem [28]. In the context of the study, "policy adoption" refers to the decision to add or reject the addition of a medicine to a NML. Policy implementation is about the series of activities and processes involved when governments and other actors attempt to translate the intention of the policy to concrete action and outcomes. This can involve establishing procedures, developing guidance, transferring human and financial resources, and putting in place administrative, regulatory, and other types of supportive structures [28,29,32].

### Search strategy

We used the following search strategy: ("WHO EML" or "World health organi#ation EML" or "Essential Medicines list*" or "model list* of essential medicines" or "WHO Model List*" or "World health organi#ation Model List*" or "WHO EDL" or "World health organi#ation EDL" or "WHO Essential Drugs list*" or "World health organi#ation Essential Drugs list*" or "model list* of essential drugs" or "essential medicines program*" or ("essential medicines" adj4 (WHO or "World health organi#ation"))).tw,kf. The following databases were chosen based on the searches of similar systematic reviews: MEDLINE, Embase, Cinahal (EBSCO), Web of Science, Scopus, Cochrane, Epistemonikos, Trip, PROSPERO, and African Index Medicus. The search of databases was led by an information specialist at the Norwegian Institute of Public Health. Due to time and capacity constraints, searches were limited to published, peer-reviewed literature.

We included studies that had an explicit focus on examining the translation of WHO EML to a national setting, which we defined to involve some form of national response to the global

**Table 1. Inclusion and exclusion criteria.**

| Criterion | Inclusion | Exclusion |
|---|---|---|
| Time | 1978 to October 10, 2021 | |
| Language | English | Non-English studies |
| Type of article | Primary qualitative studies or mixed method articles that included qualitative data collection | Quantitative studies, conference abstracts, commentaries or protocols, scoping, or systematic reviews |
| Study focus | Articles that discussed factors influencing adoption and implementation of WHO EML/NML policies at a country level (that also mentioned WHO EML) | All other articles related to WHO EML studies that discussed essential medicines, medicine coverage, access to medicines, and essential medicines policy without a focus on WHO EML adaption and implementation |
| Population | All | |

WHO EML, World Health Organization Model List of Essential Medicines.

essential medicines list and that could shed light on the adaptation and implementation at the country level (Table 1). Studies published in languages other than English were excluded as we did not have the capacity within the team to extract data from these studies; see S1 Text for a list of full texts reviewed and reasons for exclusion. We did not include gray literature as we believe that the peer-reviewed studies identified in our search were sufficient to respond to our research question. Our search criteria identified 1,627 records (1,567 unique citations). In the first stage of the review process, 2 reviewers independently screened all titles and abstracts. Any disagreements were arbitrated by a third reviewer. During the second stage, each study was read in full and assessed for inclusion separately by 2 reviewers to address potential interrater differences. Final inclusion of articles was determined through consultation among all 4 authors.

## Data extraction

Two reviewers used standardized extraction forms to independently extract the authors' interpretation of findings related to the implementation of WHO EML and illustrative examples of qualitative data from the article. Where the authors' interpretation was not supported by an illustration, we labeled this finding as "unsupported," which was subsequently included in the assessment of the confidence of review findings. Quality was assessed using the Critical Appraisal Skills Programme (CASP) quality assessment tool for qualitative studies [33]. One reviewer was randomly selected to complete a quality assessment of the articles during the data extraction phase, and a second reviewer would peer review the CASP assessment (included in the Grading of Recommendations Assessment, Development and Evaluation working group-Confidence in Evidence from Reviews of Qualitative research [GRADE-CERQual] assessment; see Table 2).

## Data analysis

Findings were analyzed individually and then through collaborative interpretation among reviewers. Data were initially organized using the "SURE checklist for understanding the barriers and facilitators to implementing a policy option" [34]. The qualitative findings populated in the Supporting the Use of Research Evidence (SURE) framework were, through an iterative process, analyzed for key themes based on our research question. Extracted data and the CERQual assessments are available as S1 and S2 Data.

## Quality assessment

We assessed the confidence in the evidence findings using the GRADE-CERQual. Two reviewers completed this assessment to identify the methodological limitations, coherence, adequacy,

**Table 2. Summary of findings table: GRADE-CERQual.**

**Objective:** To identify factors affecting adaptation and implementation of WHO Model List of Essential Medicines: a qualitative evidence synthesis
**Perspective:** Decision-makers, individuals involved in NML updates
**Included studies:** Primary qualitative research

| Review finding | GRADE-CERQual assessment of confidence in the evidence | Explanation of GRADE-CERQual assessment | Studies contributing to the review finding |
|---|---|---|---|
| 1. National policymakers recognized that the process for the establishment of national medicine selection technical committees should be consultative to facilitate genuine involvement of relevant stakeholders, and committee members could have clear roles | Moderate confidence | Moderate concerns regarding methodological limitations, No/very minor concerns regarding coherence, Minor concerns regarding adequacy, and Minor to moderate concerns regarding relevance | [16,37,38] |
| 2. Decision-makers noted that gaps in leadership, such as coordination of committee discussions, weak institutional capacity, as well as limited oversight, monitoring, and evaluation are factors that impede the implementation of NMLs | Moderate confidence | Minor to very moderate concerns regarding methodological limitations, No/very minor concerns regarding coherence, Minor/moderate concerns regarding adequacy, and Moderate concerns regarding relevance | [16,20,22,39–41] |
| 3. Technical advisory committees could have a wider mandate or longer-term role, with one study from Ghana suggesting that the committees could advocate for greater adherence to NMLs | Moderate confidence | Minor to moderate concerns regarding methodological limitations, No/very minor concerns regarding coherence, No/very minor concerns regarding adequacy, and Minor concerns regarding relevance | [16,18,41] |
| 4. Enforcement strategies and policy controls from health authorities are crucial factors affecting the prescribing and availability of medicines prioritized by NMLs | Moderate confidence | Moderate concerns regarding methodological limitations, Moderate concerns regarding coherence, Minor concerns regarding adequacy, and Minor concerns regarding relevance | [16,20,42,43] |
| 5. Inconsistency between NMLs and how they are implemented at regional or hospital level at the discretion of local doctors | Moderate confidence | Minor to moderate concerns regarding methodological limitations, No/very minor concerns regarding coherence, Minor concerns regarding adequacy, and No/very minor concerns regarding relevance | [5,41,42,44] |
| 6. The extent to which a STG facilitates local implementation of a country's NML varies | Moderate to high confidence | No/very minor concerns regarding methodological limitations, Minor concerns regarding coherence, Minor concerns regarding adequacy, and Moderate concerns regarding relevance | [5,9,16,20,22,39,41,44] |
| 7. Decision-makers perceived the essential medicines concept to be less relevant to HICs than to LMICs | Moderate to high confidence | No/very minor concerns regarding methodological limitations, No/very minor concerns regarding coherence, No/very minor concerns regarding adequacy, and Minor concerns regarding relevance | [17,46] |
| 8. A NML influences prescribing behavior but with examples of both improved and worsened prescribing practices | Moderate confidence | Moderate/significant concerns regarding methodological limitations, Minor concerns regarding coherence, Moderate concerns regarding adequacy, and Moderate concerns regarding relevance | [22,40,41,43–45] |
| 9. Many countries use key global public goods, like globally produced evidence syntheses such as WHO technical reports and Cochrane reviews, to inform their decisions | Moderate confidence | Moderate concerns regarding methodological limitations, Moderate concerns regarding coherence, Moderate concerns regarding adequacy, and Minor concerns regarding relevance | [9,18,37,39] |
| 10. A key gap countries' faced when adapting WHO EML was the lack of relevant research and local data specific to the decision context | Moderate to high confidence | Minor concerns regarding methodological limitations, No/very minor concerns regarding coherence, No/very minor concerns regarding adequacy, and Moderate concerns regarding relevance | [20,22,38] |
| 11. The cost of a medicine is an important consideration for national medicine committee selection processes | Moderate confidence | Moderate concerns regarding methodological limitations, Minor concerns regarding coherence, Minor concerns regarding adequacy, and Moderate concerns regarding relevance | [5,17,20,37,39,41,48] |

(*Continued*)

**Table 2.** (Continued)

**Objective:** To identify factors affecting adaptation and implementation of WHO Model List of Essential Medicines: a qualitative evidence synthesis
**Perspective:** Decision-makers, individuals involved in NML updates
**Included studies:** Primary qualitative research

| Review finding | GRADE-CERQual assessment of confidence in the evidence | Explanation of GRADE-CERQual assessment | Studies contributing to the review finding |
|---|---|---|---|
| 12. Context-specific economic evaluations and drug utilization evaluations are areas that can support development of NMLs | High confidence | No/very minor concerns regarding methodological limitations, No/very minor concerns regarding coherence, No/very minor concerns regarding adequacy, and No/very minor concerns regarding relevance | [16,22,37] |
| 13. NMLs can have financial implications for some private health providers as regulation of medicine selection by governments can influence the price and type of medicines sold by healthcare providers | Moderate confidence | Moderate concerns regarding methodological limitations, Minor concerns regarding coherence, Moderate concerns regarding adequacy, and Minor concerns regarding relevance | [42,44,45,49,50] |
| 14. The price, production, and procurement of medicines can be influenced by a NML. For example, the cost was perceived to be an important consideration, and the result of adding a medicine to the list could, on the one hand, mean savings through bulk purchasing of medicines or, on the other hand, can result in price inflation | Moderate confidence | Minor/moderate concerns regarding methodological limitations, Minor/moderate concerns regarding coherence, Minor/moderate concerns regarding adequacy, and Moderate concerns regarding relevance | [5,16,20,44,47,51] |
| 15. A lack of clarity between the role of medicine selection and pharmaceutical services of government authorities indicate that there is a need for improved processes to strengthen NMLs as a tool that supports availability and efficient procurement | Moderate to high confidence | No/very minor concerns regarding methodological limitations, Minor concerns regarding coherence, Minor concerns regarding adequacy, and Minor/moderate concerns regarding relevance | [16,20,22,41,52] |
| 16. NML implementation in LMICs can be dependent on donors securing financing and supply of new medicines that are added to a country's NML | Moderate confidence | Minor/moderate concerns regarding methodological limitations, No/very minor concerns regarding coherence, Minor concerns regarding adequacy, and Minor concerns regarding relevance | [16,18,39] |

GRADE-CERQual, Grading of Recommendations Assessment, Development and Evaluation working group-Confidence in Evidence from Reviews of Qualitative research; HIC, high-income country; LMIC, low- and middle-income country; NML, national medicine list; STG, standard treatment guideline; WHO EML, World Health Organization Model List of Essential Medicines.

relevance, and overall confidence in the evidence (see Table 2) [35]. Articles were included for data extraction based on inclusion criteria, regardless of study quality.

## Findings

Excluding duplicates, we identified 1,567 potentially relevant articles for title and abstract screening (Fig 1). Screening based on the full texts resulted in the final inclusion of 23 publications.

The 23 articles included in this review were from 30 countries, covering every inhabited continent. Studies were mainly from lower-middle and upper-middle income, economies, as categorized by the World Bank [36]. Two articles were from high-income economies: Australia and Canada. Table 3 presents characteristics of the studies including: country of focus, World Bank income status of the country, type of qualitative data collected, interview participants, and topical focus of the qualitative inquiry.

Following the policymaking life cycle, the results of this evidence synthesis are presented through the different phases of policy adoption and policy implementation [28,29,31]. The

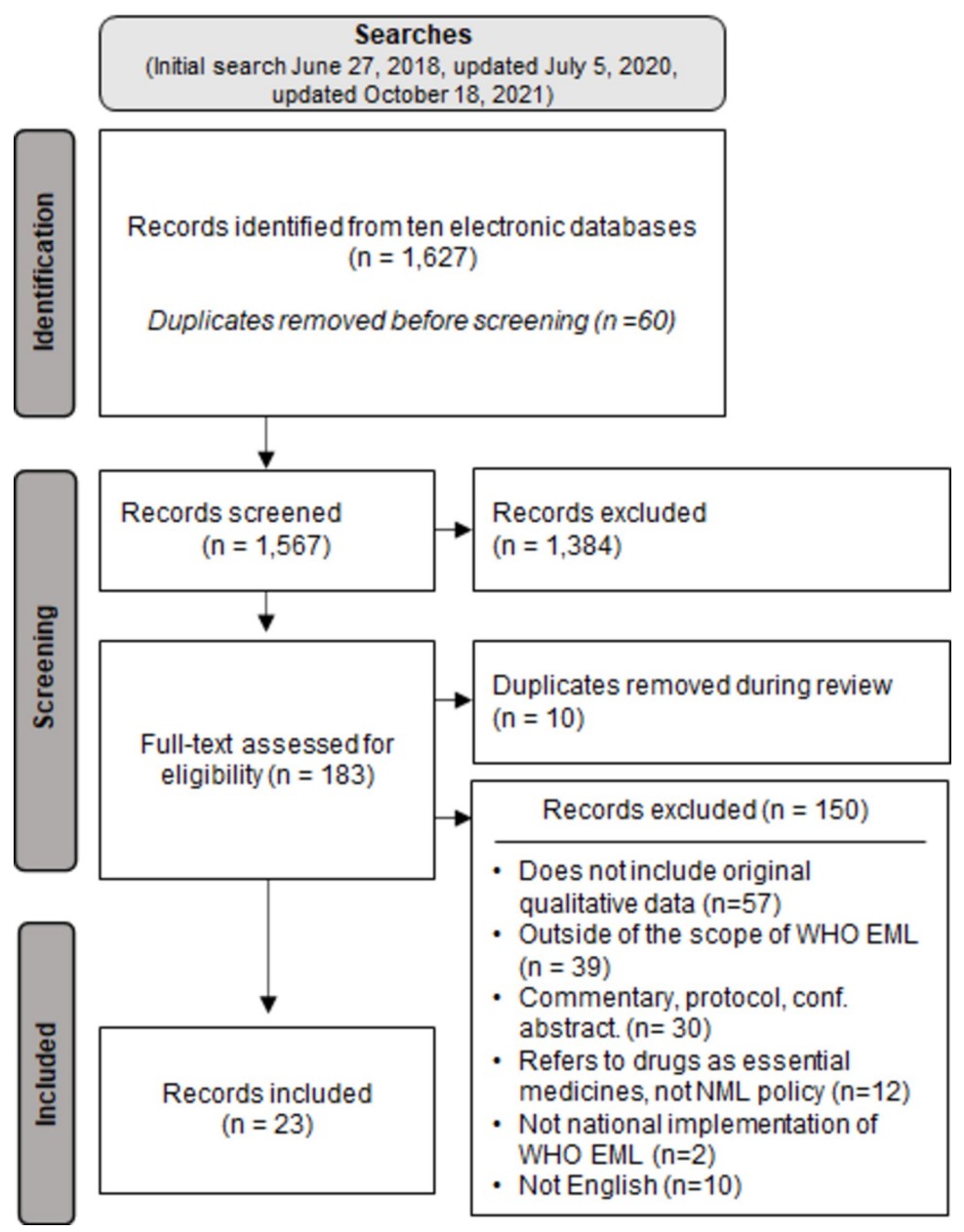

**Fig 1. PRISMA flowchart for the systematic review.**

summary of findings table provides key information concerning our review findings and the certainty of the evidence (Table 2).

## Designing institutions, governance, and leadership for national medicines lists

**Transparent and consultative national medicine selection committees can facilitate implementation.** A key institutional feature in many countries is a national medicine selection committee (alternatively known as a pharmaceutical therapeutics committee) that is responsible for determining those medicines to be included on a national list. How medicine

**Table 3. Study CHARACTERISTICS of included articles.**

| Reference | Country (World Bank income status) [36] | Type of qualitative data collection | Type of participants | Study aim |
|---|---|---|---|---|
| Albert MA, Fretheim A, Maïga D. Factors influencing the utilization of research findings by health policy-makers in a developing country: the selection of Mali's essential medicines. *Health Res Policy Syst*. 2007;5:2–2. | Mali (LI) | In-depth semistructured interviews and group discussion | National policymakers, specifically members of the national commission that selects and updates the country's list | The selection and updating of Mali's NEML |
| Atukunda EC, Brhlikova P, Agaba AG, Pollock AM. Civil Society Organizations and medicines policy change: a case study of registration, procurement, distribution and use of misoprostol in Uganda. *Soc Sci Med*. 2015;130:242–249. | Uganda (4 districts) (LI) | Policy documents, procurement data, and 82 key informant interviews | Interviews with government officials, healthcare providers, and CSOs in 4 Ugandan districts including Kampala, Mbarara, Apac, and Bundibugyo between 2010 and 2013 | The role of CSOs in promoting access to medicines |
| Bailey MCAAA, Galea G, Rotem A. From policy to action: access to essential drugs for the treatment of hypertension in the Small Island States (SIS) of the South Pacific. *Pac Health Dialog*. 2001;8(1). | Cook Islands (HI), Fiji (UMI), Kiribati (LMI), Marshall Islands (UMI), Nauru (HI), Niue (HI), and Tuvalu (UMI) | Interviews with numerous people while visiting small island states of the South Pacific | Personnel in the customs, health, foreign affairs, finance, commerce, and industry departments/ministries | To understand bulk purchasing and pooled procurement of medicines of several south pacific states |
| Brhlikova P, Maigetter K, Murison J, Agaba AG, Tusiimire J, Pollock AM. Registration and local production of essential medicines in Uganda. *J Pharm Policy Pract*. 2020;13(1). | Uganda (LI) | Interviews and document analysis | Regulators, ministry of health representatives, donors, and pharmaceutical producers between 2011 and 2015 | To examine the registration and local production of essential medicines in Uganda and understand the registration and quality assurance issues for imported and locally produced pharmaceuticals |
| Duong M, Moles RJ, Chaar B, Chen TF, World Hospital Pharmacy Research C. Essential Medicines in a High Income Country: Essential to Whom? *PLoS ONE*. 2015;10(12): e0143654. | Australia (HI) | In-depth qualitative semistructured interviews were conducted with 32 Australian stakeholders | Diverse group of stakeholders engaged in medicines selection decision-making | To explore what constitutes an "essential" medicine and how the Essential Medicines List concept functions in a HIC context |
| Fulone I, Barberato-Filho S, dos Santos MF, Rossi Cde L, Guyatt G, Lopes LC. Essential psychiatric medicines: wrong selection, high consumption and social problems. *BMC Public Health*. 2016;16:52. | Brazil (UMI) | Interviews using a Tool used to assess the Pharmacy and Therapeutics Committees of the cities (adapted from Marques, 2006) | The director of the Departments and Pharmaceutical Assistance of the Municipal Health Secretary from 3 Brazilian cities in the State of São Paulo | To investigate the use of WHO EML as a tool to evaluate the selection process for essential psychiatric medicines covered by the public system (REMUME) |
| Haque M. Essential medicine utilization and situation in selected ten developing countries: A compendious audit. *Journal of Int Soc Prev Community Dent*. 2017;7 (4):147–160. | Bangladesh (LMI), India (LMI), Nigeria (LMI), Kenya (LMI), Brazil (UMI), Mexico (UMI), Nepal (LMI), Ethiopia (LMI), Malaysia (UMI), and South Africa (UMI) | Document review (studies included interview data) | No participants | To explore the essential drug situation in 10 selected countries |

*(Continued)*

**Table 3.** (Continued)

| Reference | Country (World Bank income status) [36] | Type of qualitative data collection | Type of participants | Study aim |
|---|---|---|---|---|
| Hoebert JM, van Dijk L, Mantel-Teeuwisse AK, Leufkens HG, Laing RO. National medicines policies–a review of the evolution and development processes. *J Pharm Policy Pract*. 2013;6:5. | Sri Lanka (LMI), Australia (HI), former Yugoslav Republic of Macedonia (UMI), and South Africa (UMI) | Case studies based on 4 examples of national medicines policy formulation processes | Three experts closely involved in the policy formulation validated the document review | The present article reviews the historical development of NMPs in general, e.g., in terms of numbers and the status of implementation across various income levels. In addition, the policy formulation process is examined in more detail with case studies from 4 countries describing the historical development in these countries |
| Jarvis JD, Murphy A, Perel P, Persaud N. Acceptability and feasibility of a national essential medicines list in Canada: a qualitative study of perceptions of decision-makers and policy stakeholders. *Can Med Assoc J*. 2019; 191(40):E1093–E1099. | Canada (HI) | Semistructured interviews | Twenty-one key stakeholders from pharmaceutical policy, across Canada from federal government and pan-Canadian organizations, provincial and territorial government, civil society, and the private sector | To explore the perspectives of decision-makers and other key stakeholders on a possible NEML in Canada and to identify factors influencing the acceptability and feasibility of such a policy during an important pharmacare policy window using a qualitative study |
| Koduah A, Asare BA, Gavor E, Gyansa-Lutterodt M, Andrews Annan E, Ofei FW. Use of evidence and negotiation in the review of national standard treatment guidelines and essential medicines list: experience from Ghana. *Health Policy Plan*. 2019;34. | Ghana (LMI) | Case study design | No participants. The authors used document review and drew on joint recollection of experiences to document the process | To document the process for updating STGs and the NML |
| Li YYC, Sufang G, Brant P, Bin L, Hipgrave D. Evaluation, in three provinces, of the introduction and impact of China's National Essential Medicines Scheme. *Bull World Health Organ*. 2013;91(3). | China (2 rural districts) (UMI) | Questionnaire was sent to 6 district health bureaux in the study areas. Three focus group discussions were held per township | National Essential Medicines Scheme staff at the province, district, township, and village levels; patients with chronic disease were also interviewed | To study on implementation and impact of the National Essential Medicines Scheme |
| Matlala M, Gous A G, Meyer J C, Godman B. Formulary management activities and practice implications among public sector hospital pharmaceutical and therapeutics committees in a South African Province. *Front Pharmacol*. 2020;11. | South Africa (UMI) | Qualitative, nonparticipatory, observational study | Members of the provincial, district, tertiary hospital, regional hospital, and district hospital pharmaceutical therapeutic committees in Gauteng Province | To describe formulary management practices in public sector hospitals in the Gauteng Province of South Africa and to recommend strategies to improve formulary management by pharmaceutical therapeutic committees |
| Moodley L, Suleman F, Perumal-Pillay VA. Perceptions from pharmaceutical stakeholders on how the pharmaceutical budget is allocated in South Africa. *J Pharm Policy Pract*. 2021;14(1). | South Africa (7 of 9 provinces) (UMI) | Semistructured interviews | Seven pharmaceutical officials | To determine how the healthcare budget is calculated for the population of South Africa and translated into pharmaceutical expenditure for medicines provision on the STGs or Essential Medicine List items |
| Mori AT, Kaale EA, Ngalesoni F, Norheim OF, Robberstad B. The role of evidence in the decision-making process of selecting essential medicines in developing countries: The case of Tanzania. *PLoS ONE*. 2014;9 (1). | Tanzania (LMI) | In-depth interviews and document review | Eighteen key informants who were involved in updating the STGs and National Essential Medicine List | To study the process of updating the STGs and National Essential Medicine List in Tanzania and to examine the criteria and the underlying evidence used in decision-making |

*(Continued)*

**Table 3.** (*Continued*)

| Reference | Country (World Bank income status) [36] | Type of qualitative data collection | Type of participants | Study aim |
|---|---|---|---|---|
| Nsabagasani X, Hansen E, Mbonye A, Ssengooba F, Muyinda H, Mugisha J, Ogwal-Okeng J. Explaining the slow transition of child-appropriate dosage formulations from the global to national level in the context of Uganda: a qualitative study. *J Pharm Policy Pract*. 2015;8(1):19. | Uganda (LI) | In-depth interviews and follow-up validation meeting | Thirty-three stakeholder representatives | To explore stakeholders' views about the relevance of the global recommendation for child-appropriate dosage formulations in the context of Uganda |
| Odoch WD, Dambisya Y, Peacocke E, Hembre BSH and Sandberg KI. The role of government agencies and other actors in influencing access to medicines in three East African countries. *Health Policy Plan*. 2021;1–10. | Kenya (LMI), Tanzania (LMI), and Uganda (LI) | In-depth interviews, document review, and follow-up validation meeting | Twenty participants (Uganda 8, Kenya 7, and Tanzania 5) from Ministries of Health, Medical Procurement Agencies, and WHO 42 documents | To examine how government agencies and other actors, including nonstate actors and international partners in Kenya, Uganda, and Tanzania participate in and influence the process of updating their NEML and making prioritized medicines available |
| Osorio-de-Castro CGS, Azeredo TB, Pepe VLE, Lopes LC, Yamauti S, Godman B, Gustafsson LL. Policy Change and the National Essential Medicines List Development Process in Brazil between 2000 and 2014: Has the Essential Medicine Concept been Abandoned? *Basic Clin Pharmacol Toxicol*. 2017;122(4). | Brazil (UMI) | Document review using sources of health policy information on processes were collected from legislation, minutes, reports and legal ordinances, rename history, and related documents produced from 2000 to 2014 | No participants | To study the efforts to develop Brazil's national essential medicine list and policy changes from 2000 to 2014 |
| Perumal-Pillay VA, Suleman F. Selection of essential medicines for South Africa–an analysis of in-depth interviews with national essential medicines list committee members. *BMC Health Serv Res*. 2017;17(1):17. | South Africa (UMI) | In-depth interviews | Past and present members of the South Africa National Essential Medicines Committee and their task teams during the period January to April 2015 | To study how decisions are taken to include or exclude medicines on the South African NEML and provides insight into the medicine selection, review, and monitoring processes over time |
| Petrova GI, Benisheva-Dimitrova TV, Mircheva JD, Usunov JI. Study on essential drugs in Bulgaria: A model list based on the WHO essential drug formulary. *J Soc Adm Pharm*. 2000;17(1):59–63. | Bulgaria (UMI) | Case study of drug manufacturing, description and prescribing practice | Two governmental distribution companies and 3 private distribution companies | To evaluate the conditions of the pharmaceutical sector for endorsement of a national essential drugs list |
| Tang B, Bodkyn C, Gupta S, Denburg A. Access to WHO Essential Medicines for Childhood Cancer Care in Trinidad and Tobago: A Health System Analysis of Barriers and Enablers. *JCO Glob Oncol*. 2020;6:67–79. | Trinidad and Tobago (HI) | Case study methods with interview and review of 70 documents | Interviews with 9 key health system stakeholders, including healthcare providers, civil servants involved in oversight of the pharmaceutical system, and national and international policymakers | To analyze barriers to and enablers of access to essential pediatric cancer medicines in Trinidad and Tobago |

(*Continued*)

**Table 3.** (Continued)

| Reference | Country (World Bank income status) [36] | Type of qualitative data collection | Type of participants | Study aim |
|---|---|---|---|---|
| Wang D, Zhang X. The selection of essential medicines in China: Progress and the way forward. *South Med Rev.* 2011;4 (1):22–28. | China (UMI) | Literature review and 17 key informant interviews were conducted | Seventeen key informants were interviewed in both China and at WHO, including technical WHO officers at WHO HQ, regional and local offices, and government officers in China (Ministry of Health, pharmacists, and physicians) | To analyze the development of China's NEML from 1979 to 2010 and to provide suggestions on how to improve essential medicines selection in China |
| Xu S, Bian C, Wang H, Li N, Wu J, Li P, Lu H. Evaluation of the implementation outcomes of the Essential Medicines System in Anhui county-level public hospitals: a before-and-after study. *BMC Health Serv Res.* 2015;15:403. | China (Anhui Province) (UMI) | Focus group interviews | The interview participants included officials from government departments, experts in healthcare and hospital management, leaders of the surveyed hospital, chiefs of the medical, pharmacy, finance, and other relevant departments of the hospital, doctors' representatives, and trained investigators in 3 selected hospitals | To examine the impact on the operation of the hospitals through implementing the NEMS in Anhui Province and put forward some improvement measures |
| Zaidi S, Bigdeli M, Aleem N, Rashidian A. Access to Essential Medicines in Pakistan: Policy and Health Systems Research Concerns. *PLoS* ONE. 2013;8 (5). | Pakistan (LMI) | Key informant interviews, review of published and gray literature and consultative prioritization in stakeholder's roundtable | Twenty-one interviews were conducted with policymakers, providers, industry, NGOs, experts, and development partners | To improve the use of evidence in medicines policies and forge integrated responses to related challenges within the health systems |

Income status of countries from the World Bank [36].

CSO, civil society organisation; HI, high-income economy; LI, low-income economy; LMI, lower-middle income economy; NEML, national essential medicines list; NGO, nongovernmental organization; NML, national medicine list; STG, standard treatment guideline; UMI, upper-middle income economy.

selection committees are formed, their management, and their decision-making processes influence the adaptation and implementation of NMLs. Three articles indicated that national policymakers recognized that the process for the establishment of national medicine selection committees should be consultative to facilitate genuine involvement of relevant stakeholders and that committee members should have clear roles [16,37,38]. For example, a study from Mali reported lack of clarity about whether it was the medical professionals or policymakers from the Ministry of Health who had overall responsibility to access relevant information for the committee's decisions [38]: "You cannot place the responsibility on each person. If you do that it is not going to get done. You have to have a specific group whose job it is to get the information" [38].

**Leadership and coordination capacity are needed to secure oversight, monitoring, and evaluation.** Decision-makers noted that gaps in leadership, such as coordination of committee discussions, weak institutional capacity, as well as limited oversight, monitoring, and evaluation are factors that impedes the implementation of NMLs [16,20,22,39–41]. Nsabagasani and colleagues studied how Uganda responded to the 2012 WHO recommendations on child-appropriate dosage formulations and identified weak institutional capacity in convening relevant stakeholders to assess how the country could move forward in adopting the recommendations [39]. In addition, technical advisory committees could have a wider mandate or longer-term role [16,18,41], with a study from Ghana suggesting the committees could have a role in advocating for greater adherence to NMLs [18]. To improve implementation, more attention

to adherence and compliance strategies could be achieved through better coordination and oversight by scientific selection committees.

**Legislative and regulatory measures can help harmonize and ensure that NMLs are implemented.** Enforcement strategies and policy controls from health authorities are crucial factors affecting the prescribing and availability of medicines prioritized by NMLs [16,20,42,43]. Key factors preventing effective implementation can be the lack of an enforceable regulatory or legislative framework supporting compliance to NMLs, including linkages to pharmaceutical services departments [41]. Odoch and colleagues found that domestic health changes such as the devolution of health services had influenced NML updates [16]. Implementation can be particularly challenging in decentralized health systems that enable autonomy for medicine selection and use within regions and hospitals, even when a national framework for essential medicines is in place [5,20,44]. Several studies described inconsistencies between NMLs and how they are implemented at a regional or hospital level at the discretion of local doctors [5,41,42,44]. The lack of oversight of monitoring and implementation of NMLs resulted in hospitals diverging from national recommendations and making their own decisions related to medicine selection provision. Moreover, a study by Zaidi and colleagues in Pakistan reported that federal stakeholders and experts expressed concern over uneven medicine policy across regions caused by too little national coordination, accountability, and drug regulation [20]. In this setting, the devolution of the powers of the Ministry of Health impacted the coordination of medicine policies across subnational levels [20].

**Regional autonomy for medicine purchasing may conflict with NMLs.** A study by Matlala and colleagues examined implications of regional decision-making. They identified that on occasions, it was necessary for clinicians to prescribe nonformulary mechanisms and defer to hospital management for the decision if nonformulary medicines were perceived to be too expensive [5]. At a service delivery level, the balance of medicines selected and ensuring access to those medicines prescribed can be a particular challenge [22,40,41,45]. Experiences from different countries underscored how the lack of implementation guidance, including monitoring and evaluation, at a subnational level can lead to unregulated adaptation or disregard for NMLs by local policymakers or prescribers [20,22]. Experience from Pakistan indicated the need for strict regulation and monitoring for adhering to the NML and frequent surveillance of both the private and public sector to assess compliance [20]. Another study highlighted the need for drug utilization evaluations to understand drug use and patient outcomes [22].

**Standard treatment guidelines play a crucial role in translating intentions of NML to clinical practice.** Another key prerequisite for NML implementation is how the medicine policy links to recommended clinical practice, which is referred to as STG in many settings. These are intended to offer guidance on standardized treatment protocols and influence clinical behavior by guiding the prescribing of medicines. A finding that was assessed to have moderate to high confidence from the CERQual assessment (Table 2) indicated that STGs to a varied extent facilitate implementation of a country's *NML* [5,9,16,20,22,39,41,44]. A study from Trinidad and Tobago highlighted the value of national treatment protocols to harmonize the use of cancer medicines for adults that had been added to the essential medicines list and proposed that such an approach could also be applied to children's cancer medicines. The healthcare provider interviewed discussed the addition of ifosfamide, topotecan, and irinotecan to the NML after clinicians had written to the Drug Advisory Committee [9]. An observation study of pharmaceutical therapeutic committee meetings in Gauteng Province in South Africa cited examples of clinicians needing to develop guidelines for the control and use of nonformulary drugs [5]. Another study from Ghana described that despite the intention of

health officials to use an STG and a NML to guide healthcare providers in medicine selection, there was a lack of information about how they are used in clinical practice [18].

**Formal links to medicines reimbursement decisions is critical to the policy salience and NML's effects on prescribing, which may differ between HICs and LMICs.** The salience of NMLs to national policy settings is deeply intertwined with its relationship to medicine reimbursement decisions. A finding with moderate to high confidence (Table 2) was the lack of a shared understanding of what the concept of "essential medicines" precisely entail for their populations and that decision-makers in HICs perceived the essential medicines concept to be less relevant to their countries than to LMICs [17,46]. A partial explanation was because reimbursement decisions, and not revisions of an essential medicines list, determine the national or local formulary and the prioritization of medicines in national healthcare systems of most high-income countries (HICs). However, reimbursement processes are increasingly critical in LMICs with respect to achieving UHC, which means financial risk protections and prevent the patient from incurring high out-of-pocket expenses for essential medicines. Several papers in our review described the high out-of-pocket costs for patients, regardless of whether medicines are listed on NMLs [20,22,39,45,47].

We also identified that a NML can improve or worsen prescribing practices [22,40,41,43–45]. A Bulgarian study reported limited impact on prescribing behavior, because no formal body had approved the NML [43]. Other papers discussed the lack of dissemination as a reason for divergence [22,45] and the misalignment of actual use with forecasts when clinicians prescribed outside of the NML [41].

## Implementation is influenced by the capacity to manage evidence to inform NML updates, including processes for contextualizing the global evidence, utilizing local data and expert knowledge, and assessing costs

**Locally relevant updates of NMLs requires balancing global evidence, expert knowledge, and local data.** Differences between WHO EML and NMLs are expected given the diversity of countries' health challenges and resources. Ideally, countries have a wide range of tools at their disposal for securing sound adaptation, including global evidence, local specialist clinical expertise and experience, and local data on key criteria (e.g., efficacy, safety, availability, and affordability) that can be used to guide decisions about inclusion. Many countries use key global public goods, like globally produced evidence syntheses such as WHO technical reports and Cochrane reviews, to inform their decisions [9,18,37,39]. Globally produced evidence syntheses were key, for example, in Ghana, where the national medicines selection committee was supported by an evidence summaries group trained in the retrieval, appraisal, and interpretation of systematic reviews [18]. Moreover, WHO National Program Officer for Essential Drugs and Medicines was an important source of guidance about global best practices and lessons from other health systems [18].

A key gap facing countries when adapting WHO EML was the lack of relevant research and local data specific to the decision context [20,22,38], a finding that was assessed with CERQual to have moderate to high confidence. In a study from South Africa, Perumal-Pillay and colleagues identified that past and present members of the South African National Essential Medicines List Committee expressed concern that locally relevant data were lacking and that pharmacoeconomic studies were seldom directly applicable to their environment [22].

Local expert knowledge is crucial for adapting global recommendations domestically. However, studies reported the challenges of striking the right balance between evidence generation and a reliance on expert opinion [5,38,39,44]. Those deciding on which medicines to include, such as national policymakers or members of national committees, may perceive research

evidence to have limited relevance and applicability. For example, policymakers involved in the update of the EML in Mali indicated that most research was not applicable to policymaking and wanted greater involvement in shaping research on essential medicines [38]. In a study related to pediatric medicines, a roundtable discussion about prioritized policy and research concerns flagged that there was limited research related to the surveillance of the impact of the NML policy, including consequences of devolution of decision-making, and medicine availability and quality [20]. The hospital formulary management committee decision-making processes described by Matlala and colleagues highlighted the importance of clinicians sharing their clinical experience with the use of medicines that were not currently on WHO EML [5].

**There is insufficient utilization of cost-effectiveness information to manage cost considerations.** Several papers describe how the cost of a medicine is an important consideration for national medicine committee selection processes [5,17,20,37,39,41,48]. Recent additions to WHO EML have been high-cost medicines with potential to impose a significant strain on government health budgets. These drugs underscore the need for countries to consider the cost and financial implications of adding new essential medicines to NMLs [12]. In Brazil, cost has always been an important consideration, and additions of several high-priced cancer medicines have strained public and private hospital budgets [45,48]. A study from Tanzania found that there was disagreement between clinicians and pharmacists about the cost of medicines, with pharmacists expressing concern about budget implications [37]. A review finding with high confidence was the opportunity to use health economic methods, such as context-specific economic evaluations and drug utilization evaluations to support development of NMLs [16,22,37]. Several studies indicated the value of using Health Technology Assessment (HTA) as a tool to make medicines more affordable by evaluating the comparative effectiveness of drugs and to monitor and evaluate NMLs and decision-making processes for medicine selection [5,16,22]. However, they indicated a lack of access to the relevant data [16] or health economic expertise within country as a limitation to ensuring this [5,22].

## NMLs influence purchasing and prescribing by altering provider incentives, through linkages to systems for financing, procurement, and donor influence

**NMLs may influence provider incentives and drug-related incomes.** NMLs can have financial implications for some private health providers as regulation of medicine selection by governments can influence the price and type of medicines sold by providers [42,44,45,49,50]. For example, studies from China suggested that the implementation of NMLs had shifted the cost of medicines to hospitals. A study from Anhui Province county-level public hospitals described how the implementation of the NML meant that the hospitals lost an income source (so-called "drug price addition revenue") and that other hospitals had indicated that income was insufficient to make up for losses [42].

**To translate into availability and access, NML revisions must be linked to systems for financing and procurement of pharmaceuticals.** Our findings indicate that the price, production, and procurement of medicines can be influenced by a NML [20,44,51]. Matlala and colleagues described the processes used by committees in South Africa to align the NML with formulary procurement and associated cost implications [5]. Prescribing outside of EML is possible, and the nonadherence to NMLs has cost implications for the health budgets [41]. Additionally, the result of adding a medicine to the list could, on the one hand, mean savings through bulk purchasing of medicines [51]. On the other hand, this can result in price inflation [20,44].

In China, by linking essential medicines lists to policies that address the financing of pharmaceuticals, they have sought to reduce cost and increase efficient prescribing [44]. For example, the National Essential Medicines Scheme involved linking 3 key policies to their National Drugs List for primary healthcare institutions. First, a "zero mark-up policy" for essential medicines prescribed in primary healthcare institutions, to eliminate financial incentives for over-prescribing drugs. Second, making medicines on the National Essential Drugs List reimbursable by health insurance schemes, thereby transferring the financing of these drugs from out-of-pocket expenses by patients to mostly government-funded insurers. Finally, the Chinese National Essential Medicines Scheme required collective provincial bidding and procurement for medicines on the list [44].

Brhlikova and colleagues found that in Uganda, following increases to government funding for essential medicines, demand for locally produced medicines also increased [47]. While adding drugs to a NML is a sign of increased priority by the government, pharmaceutical manufacturers may not necessarily respond unless other factors, such as the potential for profits, are addressed. A study from Pakistan found that locally based manufacturers had a decreasing interest in manufacturing "orphan drugs" (e.g., drugs for neglected or rare diseases), despite the Ministry of Health mandating the production of these drugs [20]. In this instance, the key factor was the limited profitability of producing these medicines.

A lack of clarity between the role of medicine selection and pharmaceutical services of government authorities, a finding that was considered to have moderate to high confidence, indicated that there is a need to strengthen NMLs as a tool that supports availability and efficient procurement [16,20,22,41,52]. Inclusion in a NML based on efficacy and cost is, by itself, insufficient to promote access. A study from Brazil highlighted that there was limited interaction between the processes for selecting medicines and the processes for procuring them. The study described how a medicine would be selected based on efficacy and cost without considering how widely available it was in terms of pharmaceutical procurement. This negatively affects patients' health outcomes, as medicines may be prescribed, but lack of access means that they lapse in supply and result in incomplete treatment [22]. In Trinidad and Tobago, when updating WHO Children's List, poor communication and siloed paper-based information systems between agencies involved in children's access to medicines was cited as a challenge to implementation [9].

**Reliance on donors can influence additions to NMLs and subsequent procurement and supply.** An external factor affecting NML implementation in LMICs was dependence on donors in securing financing and supply of new medicines that are added to a country's NML [16,18,39]. In Uganda, policymakers struggled to respond to global EML changes recommending child-appropriate dosage formulations due to the additional cost implications of these medicines and expressed reliance on global financing from the Gates Foundation and other donors such as the Global Fund to Fight AIDS, Tuberculosis and Malaria [39]. Moreover, donors in Uganda were operating outside of the national policy environment and distributing medicines at the community level through vertical health programs [39]. In their study from Uganda, Brhlikova and colleagues reported that donor procurement policies were hindering local production of medicines [47].

## Discussion

WHO EML has played a critical role in guiding the country-level selection and financing of medicines for more than 4 decades. Accordingly, countries have accumulated a wealth of experience in utilizing WHO EML as a global normative tool and adapting it to the national context. Countries have also responded to its evolving normative value over time, with the

inclusion of comparatively expensive and patented antiretroviral medicines in 2002 marking a pivotal moment in WHO EML's history [1]. In recent years, the inclusion of high-priced medicines, such as for cancer and hepatitis C [16,53], have seen WHO's Expert Committee on essential medicines using WHO EML proactively to promote greater affordability and access [54]. Attention has also focused on the functioning of WHO EML within the larger goal of health systems strengthening [7].

To our knowledge, this is the first systematic review of qualitative country-level evidence to identify key factors influencing the adaptation and implementation of WHO EML at the national level. The crucial role country-level institutional structures play in implementing essential medicines policies have been highlighted by high-level reviews, such as the 2017 Lancet Commission on essential medicines for UHC [55]. To translate global revisions to meaningful national medicines policy, our review identified a wide range of institutional features that countries need. Medicine selection committees that are transparently managed and with clear roles and responsibilities can encourage an effective adaptation process by securing the involvement of relevant expertise and stakeholders. Moreover, to improve implementation of NMLs, more attention to adherence and compliance strategies could be achieved through better coordination and oversight by these committees or other types of institutions that are delegated the necessary authority. National policymakers may—especially in decentralized health systems—need legislative and regulatory frameworks that can secure adherence to the NML and harmonize implementation across subnational levels and clinical entities (e.g., hospitals or regional health trusts) that are responsible for procuring medicines. Finally, once new medicines are prioritized for inclusion, formulation of STGs and equivalent guidance are critical to translating changes from the NML to clinical practice and value for patients.

One barrier to the process of updating a NML in LMICs is the cost. This is especially true when it is necessary for selection committee members to be paid daily sitting allowances, as well as additional fees for consultation meetings and dissemination of the list [18,39]. Many countries—especially those where the health sector and academic institutions are underresourced—may lack resources and the specialized knowledge and skills to strengthen these institutions. There is potential for international collaborative initiatives as well as international assistance with country-level presence, such as WHO country offices, to better support these institutional aspects to successfully translate global revisions to locally relevant adaptions.

Our review highlights how revisions and adaptations of a NML cannot occur in isolation from other areas of the pharmaceutical system. Greater clarity is needed about the relationship between a NML and the larger systems for regulating, financing, procuring, and delivering pharmaceuticals. Opaque systems and lack of formal links between the shaping of a NML and the financing and procurement of medicines can impede the translation of NMLs to available medicines in clinical practice. Provider incentives—if misaligned and left uncorrected—can impede the goals of an essential medicines policy [56,57]. Moreover, medicine programs led by international partners or donors may undermine government policy by, for example, offering medicines that are not listed on NMLs and not funded through national insurance schemes. This may, in turn, be detrimental to the value NMLs have as a tool for fair and progressive realization of UHC. A key difference between many HICs and LMICs with respect to linking essential medicines policy to the broader pharmaceutical system is the use of reimbursement lists with tight links to national health policy and financing of pharmaceuticals [30,46,58].

Finally, the review identified the relationship between revisions to NMLs and the use of HTAs to be underdeveloped. Since 2015, successive versions of WHO EML have seen additions of patented high-cost medicines [16,53]. These drugs have provoked the access to medicines debate in HICs and reinforced the idea that high-cost drugs are a challenge for all health

systems, regardless of a country's economic status [16,46]. Moreover, these additions have challenged the affordability of essential medicines which is defined as: ". . . those [medicines] that satisfy the priority health care needs of the population. . . at a price the individual and the community can afford" [59]. The recent expansions of WHO EML increase the necessity, especially in countries that face considerable resource scarcity and where medicines continue to account for a large proportion of health spending [60], to have sound prioritization processes. These should be guided by evidence on disease prevalence, efficacy and safety, and comparative cost-effectiveness. Increasingly, countries are using HTA to inform their priority-setting decisions. However, the full potential of health economics and HTA to manage government health spending is yet to be optimized during the decision-making processes of national medicine selection committees [20,22,39,49]. Our review identified that there are gaps in local data and domestic expertise in the context of implementation of NMLs, representing a barrier to locally relevant HTA and related prioritization processes [20,22,38]. Further guidance, capacity development, and collaboration on designing comprehensive prioritization processes that use HTA and incorporate local evidence to inform national medicines policy is needed. Such processes should also influence the decisions of international partners and donors that finance medicine programs.

To gain greater insight into country-level implementation of WHO EML, there is a need for further primary country-level research on managing implementation processes of medicine policies. This includes the role of devolution as well as how the lists are disseminated to prescribers after inclusion of new medicines.

## Limitations

Studies were excluded if they were not in English or if they did not include qualitative data that documented experiences and perceptions of relevant stakeholders through interview or document review. There is therefore a risk that important experiences and perspectives published in gray literature may have been missed in this review. An information specialist completed the search 3 times, in June 2018, July 2020, and October 2021.

## Conclusions

The findings of this study may be valuable for national policymakers and practitioners, who are developing and implementing global normative guidance on essential medicines. This qualitative evidence synthesis documented a complex web of actors involved in adapting and implementing an EML, including clinicians, pharmacists, hospital administrators, insurance providers, national policymakers, the pharmaceutical industry, and international donors. These actors engage in a wide range of processes influencing implementation: policymaking, production, procurement, purchasing, prescribing, adherence, compliance, and enforcement. Overall, these actors and processes underscore the complexity and interdependencies inherent to implementation. To maximize the value of NMLs, greater investments should be made in different types of institutions that are needed to support various stages along the implementation pathway from global norms to adjusting prescriber behavior. Further research on linkages between NMLs, procurement, and the availability of medicines will provide additional insight into optimal implementation.

## Supporting information

**S1 Checklist. PRISMA Checklist.** PRISMA, Preferred Reporting Items for Systematic Reviews and Meta-Analyses.
(DOCX)

**S1 Text. List of full texts reviewed and reasons for exclusion.**
(DOCX)

**S1 Data. EML CERQual assessment of individual findings.** CERQual, Confidence in Evidence from Reviews of Qualitative research; EML, Model List of Essential Medicines.
(XLSX)

**S2 Data. Data extraction QES WHO EML.** QES, qualitative evidence synthesis; WHO EML, World Health Organization Model List of Essential Medicines.
(XLSX)

## Acknowledgments

We would like to thank the following people for their contribution: Astrid Merete Nøstberg, information specialist at the Norwegian Institute of Public Health, and Simon Lewin and Lumbwe Chola who provided guidance on the methods and manuscript.

## Author Contributions

**Conceptualization:** Elizabeth F. Peacocke, Sonja L. Myhre, Unni Gopinathan.

**Data curation:** Elizabeth F. Peacocke, Sonja L. Myhre, Hakan Safaralilo Foss.

**Formal analysis:** Elizabeth F. Peacocke, Sonja L. Myhre, Hakan Safaralilo Foss, Unni Gopinathan.

**Methodology:** Elizabeth F. Peacocke, Sonja L. Myhre, Hakan Safaralilo Foss.

**Project administration:** Elizabeth F. Peacocke.

**Supervision:** Unni Gopinathan.

**Validation:** Elizabeth F. Peacocke.

**Writing – original draft:** Elizabeth F. Peacocke, Unni Gopinathan.

**Writing – review & editing:** Elizabeth F. Peacocke, Sonja L. Myhre, Hakan Safaralilo Foss, Unni Gopinathan.

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
