## [Editor Report · Decision Letter 0]

27 Jul 2021

Dear Dr Peacocke, 

Thank you for submitting your manuscript entitled "National adaptation and implementation of the WHO Model List of Essential Medicines: A qualitative evidence synthesis" for consideration by PLOS Medicine.

Your manuscript has now been evaluated by the PLOS Medicine editorial staff and I am writing to let you know that we would like to send your submission out for external peer review.

Please re-submit your manuscript within two working days, i.e. by Jul 29 2021 11:59PM.

Kind regards,

Beryne Odeny

Associate Editor

PLOS Medicine

---

## [Decision Letter · Decision Letter 1]

22 Sep 2021

Dear Dr. Peacocke,

Thank you very much for submitting your manuscript "National adaptation and implementation of the WHO Model List of Essential Medicines: A qualitative evidence synthesis" (PMEDICINE-D-21-03040R1) for consideration at PLOS Medicine. 

[LINK]

In light of these reviews, I am afraid that we will not be able to accept the manuscript for publication in the journal in its current form, but we would like to consider a revised version that addresses the reviewers' and editors' comments. Obviously we cannot make any decision about publication until we have seen the revised manuscript and your response, and we plan to seek re-review by one or more of the reviewers. 

We expect to receive your revised manuscript by Oct 13 2021 11:59PM. Please email us (plosmedicine@plos.org) if you have any questions or concerns.

We look forward to receiving your revised manuscript. 

Sincerely,

Beryne Odeny

Associate Editor 

PLOS Medicine

plosmedicine.org

Before we proceed, please respond to the following:

1) Abstract:

a) Please report your abstract according to PRISMA for abstracts, following the PLOS Medicine abstract structure (Background, Methods and Findings, Conclusions) http://www.plosmedicine.org/article/info:doi/10.1371/journal.pmed.1001419 .

b) Please combine the Methods and Results sections into one section, “Methods and findings”. 

c) Please ensure that all numbers presented in the abstract are present and identical to numbers presented in the main manuscript text.

d) In the last sentence of the Abstract Methods and Findings section, please describe the main limitation(s) of the study's methodology.

3) Please submit a completed PRISMA checklist. We understand that not all items on the checklist will be directly relevant, and these can be marked as ‘not applicable’. 

a) The PRISMA guidelines provided at the EQUATOR site http://www.equator-network.org/reporting-guidelines/prisma/

c) Please add the following statement, or similar, to the Methods: "This study is reported as per the Preferred Reporting Items for Systematic Reviews and Meta-Analyses (PRISMA) guideline (S1 Checklist)."

4) In line with PLOS Medicine’s guidelines, please update your search to the present time and provide the beginning and end dates of your search.

5) Please avoid assertions of primacy (e.g.," This is the first systematic review of…"). Please temper claims of primacy of results by stating, "to our knowledge" or something similar.

6) References: 

a) Please reformat the citation style into PLOS Medicine's format. Please ensure there is no space between numbers in reference call outs within the main text, and the period is placed after the call out. For example, “…NLMs [7,8].”

b) Please ensure that journal name abbreviations consistently match those found in the National Center for Biotechnology Information (NCBI) databases. https://journals.plos.org/plosmedicine/s/submission-guidelines#loc-references. 

c) Please update reference #48 or remove if it has not yet been published

d) Please provide weblinks and access dates for reference # 2, 3, 4 and 43. In references #4 ad 43 please update the author’s name to WHO. 

7) Please remove the ‘Funding’, from the end of the main text. In the event of publication, this information will be published as metadata based on your responses to the submission form.

Comments from the reviewers:

Reviewer #1: The paper claims to be the first systematic review of qualitative country-level evidence to identify key factors influencing the adaptation and implementation of WHO EML at national level. It brings comprehensive review of what is known (published) and unknown on barriers to implementation of WHO EML concept.

The claims are properly placed in the context of previous literature, and the authors have treated the literature fairly.

The data and the analyses fully support the claims. It would be very useful if authors provided the list (as a supplementary document) of excluded papers with the rationale for the exclusion.

Pg 5. It would be nice if authors checked income status with the World Bank and clearly state it in e.g. Table 2. More elaborate discussion of findings in light of income status of the country in question would be welcome? If data are not available on how many participants were in particular studies, maybe to state that fact, as an asterix in the table?

Pg 4., paragraph #3, "…, Two reviewers…" the letter T should not be capital.

Protocol of this systematic review has been registered at PROSPERO, and there are no major deviations from it. Methods are appropriately selected to answer the research objective.

This paper identifies gaps in EML concept implementation and provides valuable information necessary for better implementation of EML concepts.

The authors have selected GRADECERQual approach which is more suitable for analysis of this kind of qualitative data than the methods Equator Network recommends: SPQR and COREQ, since the latter two are more suitable for classic qualitative research - interviews.

The details of the methodology are sufficiently described and allow experiment to be reproduced.

The manuscript is well organized and written clearly enough to be accessible to non-specialists.

Reviewer #2: This systematic review was performed according to standard methodology and was well written. The author derive some relevant points from the literature. I had a few points to consider:

1. Why did the authors not include quantitative articles in the review? The qualitative lessons would be a lot more impactful if readers understood if there was any empirical basis on which the lessons were based. On a more minor note, the authors provide only in the table a quick summary of the underlying basis on which the recommendations were made in the qualitative articles themselves; it would be useful for some of that information to be integrated into the text as well.

2. The authors appear to consider all of the articles as contributing equally to the recommendations, but the articles were published across 2 decades and across many different countries from HIC to LMIC. Is an article from 20 years ago conducted in possibly a much different local health environment relevant to recommendations today? More background on the context in which the recommendations were made would be helpful -- i.e. how can policymakers in France consider lessons from a study from China if their implementation systems are very different? Even on a more specific note, the authors assigned the articles to different levels of confidence (high->low) based on the features of the report, but then appeared not to use those assignments in deriving their recommendations, since they appear to equally rely on low confidence conclusions from citations #29 and #32 just as strongly as high-confidence citations. Some recognition that a recommendation is based on low-confidence insights would be helpful. The same comment refers to the different qualitative bases on which the conclusions in the articles are reached -- some are based on actual qualitative research, some just on document review -- which is more reliable?

3. Greater detail in the discussion should be provided on how to implement some of these ideas, perhaps identifying specific organizations or prioritizing recommendations with different levels of urgency.

4. Minor stylistic note: There are too many acronyms, which makes for complicated reading.

Reviewer #3: This study addresses an important topic, namely, the adaptation and implementation of the WHO list of essential medicines. This is an interesting topic and may be applicable to countries interested in adapting the WHO list. There are several areas the limit the usefulness of this study. 

# Value of WHO list of medicines

- The study has a laudatory perspective on the WHO EML, which offers a "trusted and objective reference list..." (Introduction). However, the WHO EML has several limitations and does not always list medicines that "best satisfy the priority health care needs of a country's population". In fact, the WHO EML omits many medicines for chronic medical conditions (e.g., diabetes, heart disease), which are routinely used in high-income countries. Therefore, the WHO EML might not be a reference document for many countries because it is not always grounded in the best evidence. 

- In many countries (HICs, and LMICs; e.g., India), health policy and medicines fall under regional (e.g., state, provincial, etc.) jurisdiction. There may be a national EML and additional regional lists. In such instances, the regional list takes precedence over the national list. 

- Given the above points, I am not convinced that many countries are trying to adapt the WHO EML, which raises questions on the utility/value/contribution of this study. I am not confident that this is a major problem or barrier for many countries. 

# Data Source and Search Strategy

- The database search was led by an information specialist, which is a strong point. 

- However, for this type of study, the grey literature is important because many policy documents might not be published in peer-reviewed journals. 

- In conducting a literature search, I identified articles that were not included but seemed pertinent (e.g., Jarvis et al., CMAJ 2019; essential medicines in Canada). I was not part of this study and have no vested interest, but I was curious if this article would be relevant. 

- The Limitations mention that the search was done in July 2020, and "following the last update, two studies were added for data extraction". This means, following July 2020? How were these identified? Given that the search was done in July 2020, the authors will have to consider if they should update the search because "We did not complete the search again later in 2021 as we thought it was unlikely that the addition of further studies would influence our main findings" might not convince most reviewers or editors. While new themes may not be identified, similar experiences from other countries may strengthen their conclusions. Having been involved in several systematic reviews, I recognize the effort needed to update literature searches. The authors should consider updating it. For literature sweeps, some journals accept if a single investigator screens articles and extracts data. The authors could consider this as a half-way solution. 

# Scope of Review

- Table 2 outlines the types of studies included. It was not clear how the authors decided to combine studies on 'document reviews' and 'qualitative interviews'. If including document reviews, then it becomes more important to include the grey literature, as described above. 

- It would be helpful if the study clarified if this list pertained to the adult only or adult and pediatric list of medicines (I apologize if I missed it). 

# Audience

- The study should specify the intended audience and target the Discussion accordingly. It is not clear if this is directed toward government leaders, policy makers, academics, etc. Consequently, the Results and Discussion appear to lack focus. 

# Additional points

- Many strongly worded statements are without reference(s). Please review and provide references. For example "HICs may have more robust reimbursement processes..."

- Please review for typos

- The Abstract should be revised. Were the 10 themes identified in all 18 studies (the current version suggests that). It would be helpful to summarize if one/few themes were found in all studies or a majority of the studies. 

Overall, this is an interesting question, but the value/utility remains uncertain.

[LINK]

---

## [Decision Letter · Decision Letter 2]

3 Feb 2022

Dear Dr. Peacocke,

Thank you very much for re-submitting your manuscript "National adaptation and implementation of the WHO Model List of Essential Medicines: A qualitative evidence synthesis" (PMEDICINE-D-21-03040R2) for review by PLOS Medicine.

I have discussed the paper with my colleagues and the academic editor and it was also seen again by two reviewers. I am pleased to say that provided the remaining editorial and production issues are dealt with we are planning to accept the paper for publication in the journal.

[LINK]

We look forward to receiving the revised manuscript by Feb 10 2022 11:59PM.   

Sincerely,

Beryne Odeny, 

PLOS Medicine

plosmedicine.org

Requests from Editors:

1) Abstract: please change the sub-heading “Objective” to “Background.” Under background, please provide the context of why the study is important. The final sentence should clearly state the study question.

2) Please reformat your Author summary and use bullet points instead of paragraphs. Use no more than 4 bullet points per sub-heading.

Comments from Reviewers:

Reviewer #2: No further comments

Reviewer #3: The authors have provided adequate responses to all questions/comments. I have no further comments.

[LINK]

---

## [Editor Report · Decision Letter 3]

11 Feb 2022

Dear Dr Peacocke, 

On behalf of my colleagues and the Academic Editor, Dr. Aaron S Kesselheim, I am pleased to inform you that we have agreed to publish your manuscript "National adaptation and implementation of the WHO Model List of Essential Medicines: A qualitative evidence synthesis" (PMEDICINE-D-21-03040R3) in PLOS Medicine.

PRESS

Sincerely, 

Beryne Odeny 

PLOS Medicine